# Effects of Intestinal Bacterial Hydrogen Gas Production on Muscle Recovery following Intense Exercise in Adult Men: A Pilot Study

**DOI:** 10.3390/nu14224875

**Published:** 2022-11-18

**Authors:** Nobuhiko Eda, Saki Tsuno, Nobuhiro Nakamura, Ryota Sone, Takao Akama, Mitsuharu Matsumoto

**Affiliations:** 1Waseda Institute for Sport Sciences, Waseda University, Tokorozawa 359-1192, Japan; 2Department of Fundamental Education, Dokkyo Medical University, Tochigi 321-0293, Japan; 3Dairy Science and Technology Institute, Kyodo Milk Industry Co., Ltd., 20-1 Hirai, Hinode 190-0182, Japan; 4Faculty of Sport Sciences, Waseda University, Tokorozawa 359-1192, Japan; 5Japan Institute of Sports Sciences, Kita-ku, Tokyo 115-0056, Japan

**Keywords:** beverage, hydrogen gas, high-intensity exercise, placebo, overreaching

## Abstract

This study aimed to examine the effects of hydrogen gas (H_2_) produced by intestinal microbiota on participant conditioning to prevent intense exercise-induced damage. In this double-blind, randomized, crossover study, participants ingested H_2_-producing milk that induced intestinal bacterial H_2_ production or a placebo on the trial day, 4 h before performing an intense exercise at 75% maximal oxygen uptake for 60 min. Blood marker levels and respiratory variables were measured before, during, and after exercise. Visual analog scale scores of general and lower limb muscle soreness evaluated were 3.8- and 2.3-fold higher, respectively, on the morning after treatment than that before treatment during the placebo trial, but not during the test beverage consumption. Urinary 8-hydroxy-2′-deoxyguanosine (8-OHdG) concentrations and production rates significantly increased with placebo consumption; no changes were observed with test beverage consumption. After exercise, relative blood lactate levels with H_2_-producing milk consumption were lower than those with placebo consumption. A negative correlation was observed between the variation of 8-OHdG and the area under the curve (AUC) of breath H_2_ concentrations. Lipid oxidation AUC was 1.3-fold higher significantly with H_2_-producing milk than with placebo consumption. Conclusively, activating intestinal bacterial H_2_ production by consuming a specific beverage may be a new strategy for promoting recovery and conditioning in athletes frequently performing intense exercises.

## 1. Introduction

During exercise, oxygen consumption in active tissues noticeably increases and followed by an increase in reactive oxygen species (ROS) production [1,2]. Although ROS produced by regular moderate training have physiological roles, such as modulating cellular signaling pathways and controlling numerous redox-sensitive transcription factors, ROS overproduction by acute and strenuous bouts of aerobic exercise cause oxidative damage of lipids, proteins, and DNA [3], which leads to delayed recovery from muscle damage and fatigue, reduced muscle contractility, inflammation, and immune deficiency [4,5]. Normally, antioxidant defense systems are upregulated and the high levels of oxidative stress markers decrease with time after acute exercise; however, extreme intense exercise can induce high baseline oxidative stress markers and low plasma antioxidant capacity for more than 1 month [6]. As such, reducing or quickly eliminating the excessive ROS generated by acute intense exercise is important for athletic conditioning.

Since it was discovered in 2007 that hydrogen gas (H_2_) has selective antioxidant properties [7], multiple studies have shown that H_2_ has beneficial effects in diverse animal models and human disease. For example, H_2_ supplementation positively affects numerous biological phenomena, including inflammation, allergies, and metabolic syndrome [8]. In sports science, the antioxidative effects of H_2_ on athletes who repeatedly perform high-intensity exercise have been demonstrated [9]. The intake of H_2_ induces improvements in exercise performance [10,11] and recovery from muscle inflammation and fatigue [12,13]. Furthermore, H_2_ supplementation has been shown to enhance fatty acid metabolism [14]. 

H_2_-rich water ingestion has been the primary method of supplying H_2_ to the body. H_2_-rich water and saline can be easily administered; however, ingested H_2_ obtained through this method remains in the body for approximately 20–30 min [15,16]. When indigestible dietary fiber reaches the large intestine, the intestinal bacteria utilize it to produce H_2_, short-chain fatty acids, carbon dioxide, and other substances [17]. Subsequently, H_2_ is dispersed systemically via the portal circulation [18,19]. A beverage comprising cow milk and indigestible dietary fiber was developed to enable H_2_ production in the colon of most Japanese individuals, irrespective of personal differences in the intestinal microbiota [16]. Therefore, this study investigated the effect of H_2_-producing milk and intestinal microbiota-derived H_2_ on muscle damage, antioxidant activity, and energy metabolism recovery after high-intensity exercise.

## 2. Materials and Methods

### 2.1. Participants

Participants were provided with detailed explanations of the risks, inconveniences, and potential benefits of the study before signing an informed consent form. All participants who had completed a comprehensive medical examination within the preceding year were recruited at Waseda University for 1 month (October 2017). Participants with milk allergy, lactose intolerance, and those without intestinal H_2_ production after test beverage (H_2_-producing milk) ingestion were excluded. This study was approved by the Ethics Committee on Human Research of Waseda University (#2017-092) and was conducted in accordance with the principles of the Declaration of Helsinki.

### 2.2. Experimental Protocol

All study participants were involved in two trials. The first and second trials involved test beverage and placebo ingestions, respectively, before exercise. This study had a double-blind, randomized, crossover design (Trial registration: UMIN000029238). Participants and observers were blinded to the assignment throughout the trial period. Double-blinding was achieved by labeling the test beverage with an identification number only, and randomization was based on a table of random numbers. Participants underwent the trials at the same time of the day with intervals of >5 days in the Tokorozawa campus of Waseda University. They consumed the prescribed diets, which did not produce H_2_ during digestion, excluding test beverages, at 8:00 pm the night before the trial date and at 9:00 am, 10:30 am, and 8:00 pm on the day of the trial. No food or fluids, except mineral water, were consumed until after the measurements were performed in the morning after the trial. Participants ingested the test beverage or placebo at 9:00 am on the day of trial, 4 h before performing the exercise. They performed a pedaling exercise at 75% maximal oxygen uptake (VO_2max_) for 60 min in a climate-controlled room with a temperature of 21 °C and 50% relative humidity. During exercise, the pedaling rate was maintained at 60 revolutions per minute (rpm). Measurements were performed before, immediately after, 30 min after, and 60 min after exercise and the morning after the trial. After the completion of the trial, participants were interviewed to confirm appropriate test beverages and diet intake and the maintenance of their usual routine activities.

### 2.3. Preparation of the Test Beverage

H_2_-producing milk was prepared by adding galacto-oligosaccharide (2 g), maltitol (2 g), and glucomannan (0.2 g) together, which are active ingredients that produce intestinal H_2_ in cow’s milk, as described in our previous report [20]. Details are described in the Appendix A. These beverages had the same color and taste and were packed separately into unlabeled pots. A questionnaire was used to ascertain whether participants were aware of the allocation.

### 2.4. Determination of VO_2max_

Participants performed a pedaling exercise on a cycle ergometer (AEROBIKE75XLII; Combi Wellness, Tokyo, Japan) at least 1 week before the trial to determine the VO_2max_. After a standardized warm-up for 3 min, the protocol began at 90 W and was increased by 15 W/min until volitional exhaustion [21]. During the test, the respiratory gas levels were analyzed using a pre-calibrated automatic gas analyzer (AE-310s; Minato Medical Science, Tokyo, Japan). The heart rate (HR) was monitored continuously using an HR monitor (BSM-2401; Nihon Kohden, Tokyo, Japan). At least three of the following five criteria were used to verify the attainment of VO_2max_: oxygen consumption reaching a plateau, respiratory exchange ratio > 1.1, HR >90% of the predicted maximal value (220—age), Borg scale score >18 [22], and pedaling rate <50 rpm. 

### 2.5. Respiratory Gas Analyses

During the exercise sessions in both trials, the respiratory exchange ratio, lipid oxidation, carbohydrate oxidation, oxygen uptake (VO_2_), and ventilation (VE)/volume of exhaled carbon dioxide (VCO_2_) were measured using a pre-calibrated automatic gas analyzer (AE-310s; Minato Medical Science, Tokyo, Japan). Respiratory gas concentrations and HR were analyzed at 0, 3, 15, 30, 45, and 60 min during the exercise sessions. Energy expenditure was calculated based on the method of Livesey and Elia [23]: energy expenditure (kcal/min) = [oxygen consumption × (15.480 + 5.550 × respiratory exchange ratio)/4.184]/1000.

### 2.6. Blood Sample Measurements

Blood samples were obtained from the antecubital vein, and the blood leukocyte count was measured. Blood samples obtained in blood-collecting vessels for serum separation or containing perchloric acid were maintained for 30 min at room temperature; thereafter, the serum and deproteinized supernatants were collected from clotted blood samples after centrifugation at 3000 rpm for 15 min. Blood parameters were measured in the laboratory of BML Inc. (Tokyo, Japan) and were corrected using hematocrit values and hemoglobin concentrations to account for the effects of blood concentration [24].

### 2.7. Determination of 8-Hydroxy-2′-deoxyguanosine (8-OHdG) Concentration

The 8-OHdG concentration was measured as a primary outcome using an enzyme-linked immunosorbent assay kit (New 8-OHdG check; Japan Institute for the Control of Aging, Shizuoka, Japan). The procedural details are described in the Appendix A. 

### 2.8. Fatigue and Muscle Soreness

Feelings of fatigue, whole-body and lower limb muscle soreness were assessed using a visual analog scale (VAS) comprising of a 100 mm horizontal line. Participants were informed that the left and right ends of the scale represented “no fatigue or pain” and “most severe fatigue or pain imaginable”, respectively.

### 2.9. Breath H_2_ Concentration Measurements

The breath H_2_ concentration was measured at 0, 5, 6, and 7 h after test beverage or placebo ingestion, in accordance with a previously described method using a gas analyzer (TRIlyzer mBA-3000; TAIYO Instrument Inc., Osaka, Japan) [16]. Our previous study confirmed that the breath H_2_ concentrations of nearly all Japanese participants increased 5–7 h after beverage ingestion [16]. Majority of the studies reported the mean small intestinal transit time of food, water, and various dosage forms to be about 3 to 4 h after consumption [25]. Hence, H_2_ produced by gut microbiota from beverage ingredients that reached the colon was measured. 

### 2.10. Fecal Microbiota Analysis

The first feces defecated within 24 h after the end of the exercise were collected. Subjects used feces collection sheets “Nagaseru” (Atleta, Osaka, Japan) for fecal collection to prevent contamination with water and chemicals from the toilet. Immediately after defecation, a part of the feces (approximately 1–2 g) was collected in a tube and kept in a freezer at −80 °C until use. Intestinal microbiota was analyzed using 16S rRNA gene amplicon sequencing to identify intestinal bacteria involved in H_2_ production. The methods of fecal bacterial DNA extraction, 16S rRNA gene amplicon sequencing, data processing, and sequence alignment are described in the Appendix A.

### 2.11. Statistical Analysis

The Shapiro–Wilk normality test was used to investigate whether a variable is normally distributed in each data set. The breath H_2_ concentration at each time point was converted to the rank value and evaluated using a two-way repeated-measures analysis of variance (ANOVA) and an estimated marginal means test. The area under the curve (AUC) of the breath H_2_ concentration and other AUCs were evaluated using the Wilcoxon signed-rank test and paired *t*-test, respectively. VAS scores were converted to the rank value and analyzed using two-way repeated-measures ANOVA; they were adjusted using the Bonferroni method. Serum, urine, and breath marker levels were analyzed using a two-way repeated-measures ANOVA adjusted using Dunnett’s multiple comparison test or Bonferroni method via the Friedman test. Correlations were analyzed using Pearson’s product-moment correlation coefficient. The relative abundance ratio of fecal bacteria was determined using the Wilcoxon signed-rank or rank-sum test, which was adjusted using the Benjamini–Hochberg method. All statistical analyses, except those involving the Benjamini–Hochberg method, were performed using SPSS version 25.0.0 (IBM, Armonk, NY, USA) and R statistical software version 3.4.4 (R Foundation for Statistical Computing, Vienna, Austria). The Benjamini–Hochberg method was performed using R statistical software version 3.4.4 (R Foundation for Statistical Computing, Vienna, Austria). All differences were considered significant at *p* < 0.05. The sample size was estimated using G*Power 3 [26]. To detect the changes in the respiratory exchange ratio, blood lactate, and oxidative stress by test beverage and placebo ingestions with a power of 80% and alpha level of 5%, a sample size of ≥6 participants was required.

## 3. Results

### 3.1. Breath H_2_ Concentration Changes

Although 10 people participated in this study, we excluded the data of two participants (participants Nos. 5 and 9) whose breath H_2_ concentration AUCs with H_2_-producing milk ingestion were smaller than the maximum value of the breath H_2_ concentration AUC with placebo ingestion (participant 5, 6.55; participant 9, 17.18; maximum value with placebo, 20.31) because their data were not suitable for determining the effects of H_2_ produced by consuming the test beverage. The result showed that there were 20% non-responders to the test beverage. Data from eight men (age, 25.6 ± 3.3 years; height, 171.8 ± 4.0 cm; body mass, 63.6 ± 9.3 kg; body mass index, 21.5 ± 2.5 kg/m^2^; body fat percentage, 14.9 ± 4.6%; and VO_2max_, 49.7 ± 9.1 mL/kg/min) were analyzed. The breath H_2_ concentration and its AUC after H_2_-producing milk ingestion were significantly higher at 4–6 h after test beverage ingestion than that after placebo ingestion (Figure 1).

### 3.2. Muscle Soreness and Fatigue

The VAS scores of general (*p* < 0.01) and lower limb (*p* < 0.05) muscle soreness were significantly higher the morning after treatment than that before treatment during the placebo trial but not during the H_2_-producing milk trial (Table 1). The VAS score for fatigue the morning after treatment was significantly higher during both trials than that before treatment, and there was no significant difference between the scores during both the trials.

### 3.3. Blood Samples

Almost all serum and urine marker levels significantly increased after exercise; however, no significant interactions and differences between trials were observed (Table 1 and Appendix A). Relative lactate levels during the H_2_-producing milk trial tended to be lower than those during the placebo trial immediately after exercise (*p* = 0.070; Figure 2a). We observed a negative correlation between the relative lactate levels and the AUC of the breath H_2_ concentration (r = −0.495; *p* = 0.051; Figure 2b).

### 3.4. Urinary 8-OHdG

The urinary 8-OHdG concentration and production rate significantly increased immediately after exercise during the H_2_-producing milk and placebo trials (*p* < 0.01; Appendix A). Moreover, the relative 8-OHdG concentration (*p* < 0.05) and production rate (*p* < 0.05; Figure 2b) increased immediately after exercise only during the placebo trial. Variations in the 8-OHdG concentration (r = −0.533; *p* < 0.05) and production rate (r = −0.598; *p* < 0.05; Figure 2b) during the trials showed significant negative correlations with the AUC of the breath H_2_ concentration.

### 3.5. Respiratory Variables

There were significant main effects for time for VO_2_, HR, and VE/VCO_2_; however, no significant interactions and main effects were observed between the trials (Appendix A). In addition, there were no significant differences in the VE, VCO_2_, and VE/VO_2_ during the placebo and H_2_-producing milk trials. The respiratory exchange ratio (*p* = 0.071; Figure 3a) and carbohydrate oxidation (*p* = 0.069; Figure 3b) during the H_2_-producing milk trial at 3 min after starting the exercise tended to be lower than those during the placebo trial. In contrast, lipid oxidation during the H_2_-producing milk trial at 3 min (*p* = 0.079) and 60 min (*p* = 0.099) after starting the exercise tended to be higher than that during the placebo trial (Figure 3c). During exercise, there was a significant difference in the lipid oxidation AUCs (*p* < 0.05), but not in the carbohydrate oxidation AUCs, during the placebo and H_2_-producing milk trials (Figure 3d). There was no significant difference in energy expenditure between both the trials (Appendix A).

### 3.6. Fecal Microbiota

The principal component analyses of the fecal microbiota revealed no specific cluster for either trial (Appendix A, left). Each participant had similar fecal microbiota throughout the study period, indicating that the intestinal microbiota was not significantly altered by the trials (Appendix A, right). There was no difference in the relative abundance of each bacterial group during both the trials at the phylum to genus levels. Additionally, the relative abundance of each bacterial group was not altered by the consumption of either beverage (Appendix A). All sequence data were deposited in the DDBJ Sequence Read Archive database under the accession number DRA013790.

## 4. Discussion

In this study, we examined the effects of a functional milk beverage that induces intestinal bacterial H_2_ production on fatigue prevention after intense exercise; moreover, estimations were made on the muscle damage prevention mechanisms based on data regarding oxidative stress, blood markers, and respiratory variables.

Levels of creatine kinase and myoglobin, which are muscle damage markers [27], and the total number of blood leukocytes, which indicates the level of inflammation [28] increased by intense exercise during both trials, demonstrating that intense exercise inflicts damage to the body. H_2_ has an antioxidant effect that selectively reduces the level of ROS [7], which would have the potential to induce oxidative stress that may cause a delay in recovery after muscle damage and fatigue, reduction of muscle contractility, inflammation, and immune deficiency [5]. Although the 8-OHdG concentration and production rate significantly increased with intense exercise during the placebo trial, no increase was observed during the H_2_-producing milk trial, indicating that intestinal bacteria-derived H_2_ inhibits oxidative damage caused by intense exercise. In addition, a negative correlation between the concentration or production of 8-OHdG and the breath H_2_ concentration AUC supports this suggestion. Whether 8-OHdG can be used as an oxidative marker after exercise is controversial, since there are reports stating that 8-OHdG increases by acute bout of exercise [29] and also that it is not changed by a fairly intense bout of exercise [30]. This seems to be because the change in 8-OHdG level due to exercise varies depending on exercise mode, intensity, and duration level of the individuals. In a previous study of an acute intense exercise (60 min of cycling exercise at 70% VO_2_ peak), which is similar to this study (60 min of cycling exercise at 75% VO_2_ peak), urinary 8-OHdG was reported to increase after exercise [29]; hence, urinary 8-OHdG was used as an oxidative stress marker in this study.

By ingesting H_2_-producing milk, the participants recovered from the muscle soreness the next morning, based on VAS score findings; this indicated that consuming this beverage effectively improves muscle recovery after intense exercise. Free radicals are unquestionably produced during and after various forms of muscle contractile activity [31] and are known to result in skeletal muscle damage [32]. Animal and human studies have demonstrated that antioxidant polyphenols have anti-inflammatory effects in an exercise-induced muscle damage model [33,34]. Rats that ran until exhaustion on a sealed treadmill while inhaling an H_2_-containing mixture were shown to have reduced oxidative stress, post-exercise muscle damage, and inflammation [35]. Therefore, the antioxidant effects of the H_2_-producing milk may have contributed to muscle recovery in this study. However, several reports have shown that ROS act as signaling molecules that specifically activate redox-sensitive transcription factors, which are necessary for muscle recovery after damage [36]. Hence, the antioxidant effects on muscle recovery are controversial. The differences between the harmful and beneficial effects of ROS may depend on the level of ROS produced by exercise [4]. We speculated that the beneficial effects of muscle recovery observed in our study occurred because the intense exercise-induced ROS was reduced to a level suitable to induce the recovery of muscle damage by the antioxidant power of H_2_. However, further studies are required to clarify this.

Interestingly, reduced lactate levels and upregulated lipid oxidation were caused by H_2_-producing milk consumption, and a negative correlation between lactate levels and the breath H_2_ concentration AUC was observed. Recent studies have shown that lactate is the energy source for ATP resynthesis during continuous exercise [37]; however, an increased lactate level is thought to be an indicator of muscle fatigue. During the early stage of the exercise, glycolysis—which can produce ATP for a short time—preferentially occurs, leading to lactate production. Energy production via the mitochondria occurs when more energy is needed; carbohydrates are used more when a large amount of energy is needed during a short period, while when exercise is performed for a long period, lipids are used more [38]. Therefore, it is suggested that the energy production system preferentially utilizes lipids as an energy source with H_2_-producing milk consumption; moreover, glycolysis was downregulated, resulting in lactate production reduction. On the other hand, H_2_ produced by the intestinal microbiota may facilitate the use of lipids as an energy source. Kamimura et al. [14] reported that drinking H_2_-rich water induced *Pgc-1α* gene expression and enhanced fatty acid metabolism in mice, thereby supporting the abovementioned finding. However, the effects of consuming the H_2_-producing beverage on energy metabolism should be further studied.

To the best of our knowledge, this is the first study to investigate the effects of H_2_ produced by intestinal microbiota on the human body during a high-intensity exercise. Several studies have shown that exercise induces changes in the microbiota composition [39]. Furthermore, the metabolites of various gut microbiota are essential for improving mitochondrial oxidative stress, inflammatory response, metabolism, and energy expenditure during exercise [40]. However, during this study no significant alteration of the intestinal microbiota was observed with a single dose of the test beverage, suggesting that these effects are not dependent on the alteration of the bacterial composition of the gut.

### Limitations

This study had several limitations. First, this was a pilot study that included only eight participants, which increased the likelihood of a type II error skewing the results. Second, the participants were healthy individuals, albeit not athletes. Therefore, whether these results apply to athletes who frequently perform intense exercise is unclear. Third, recovery following exercise was measured by using only VAS and there was no difference in the objective measures. Fourth, the intestinal microbiota probably produces metabolites other than H_2_ after consuming this beverage; therefore, it is possible that those unknown metabolites may have caused these effects. Finally, we may not have been able to perfectly identify the fecal site from the whole fecal sample influenced by the single dose of test beverage. Thus, further research to identify and collect the fecal site which receives the test beverage or food, required to estimate H_2_-producing bacteria, using visible indigested food item or ingredient such as seaweed is required [41].

In conclusion, this study is the first to investigate the effects of H_2_ produced by intestinal microbiota preventing muscle fatigue caused by intense exercise. H_2_-producing milk consumption reduces muscle soreness after an intense exercise; hence, this new strategy may possibly prevent overtraining syndrome and enhance the recovery and conditioning of athletes.

## Figures and Tables

**Figure 1 nutrients-14-04875-f001:**
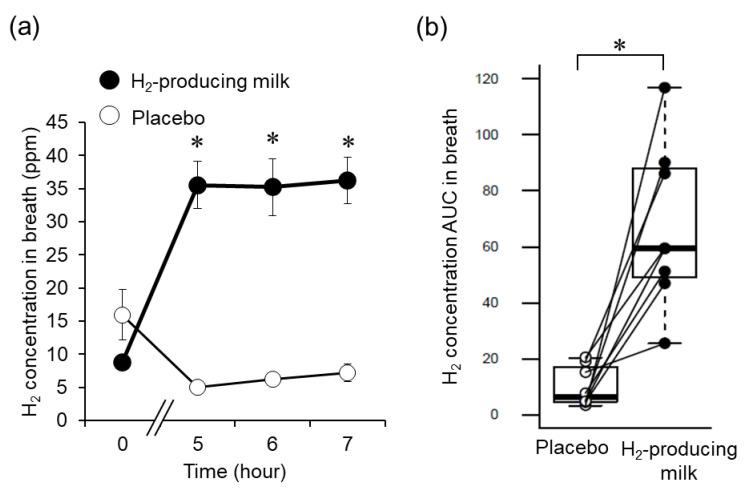
Changes in H_2_ production with the ingestion of H_2_-producing milk. (**a**) Time course of the breath H_2_ concentrations during the first 6 h after ingestion of the test beverage. Error bars represent standard error of the mean. * q < 0.05 (vs. placebo; Friedman and Wilcoxon signed-rank tests with the Benjamini–Hochberg correction). (**b**) Between-trial comparison of the area under the curve (AUC) of the breath H_2_ concentrations (0–6 h). * *p* < 0.05 (Wilcoxon signed-rank test performed after the F-test). Each part of the boxplot diagram shows the following: upper line, maximum value; lower line, minimum value; box, interquartile range; bold line inside the box, median; and white and black plots, values during the placebo and H_2_-producing milk trials, respectively.

**Figure 2 nutrients-14-04875-f002:**
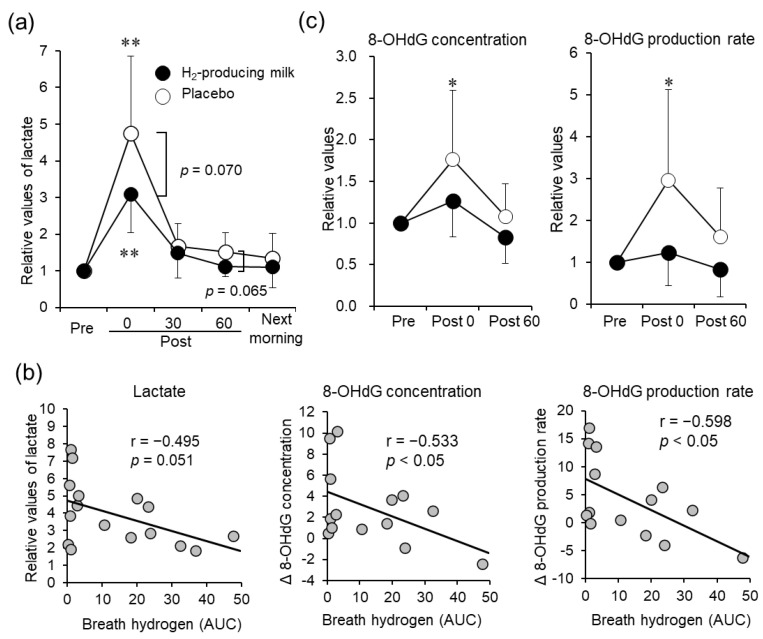
Effects of H_2_-producing milk and placebo use on blood lactate and 8-OHdG levels. (**a**) Time course of the relative lactate levels before and after exercise. (**b**) 8-OHdG concentration and production rate before and after exercise. The black and white plots represent the means during the H_2_-producing milk and placebo trials, respectively. Error bars represent the standard deviation of the mean. * *p* < 0.05, ** *p* < 0.01 (vs. before exercise; two-way repeated measures analysis of variance and Dunnett’s multiple comparison test). (**c**) Correlation between blood lactate and 8-OHdG levels and the AUC of the breath (Pearson’s product-moment correlation coefficient). 8-OHdG, 8-hydroxy-2′-deoxyguanosine; AUC, area under the curve.

**Figure 3 nutrients-14-04875-f003:**
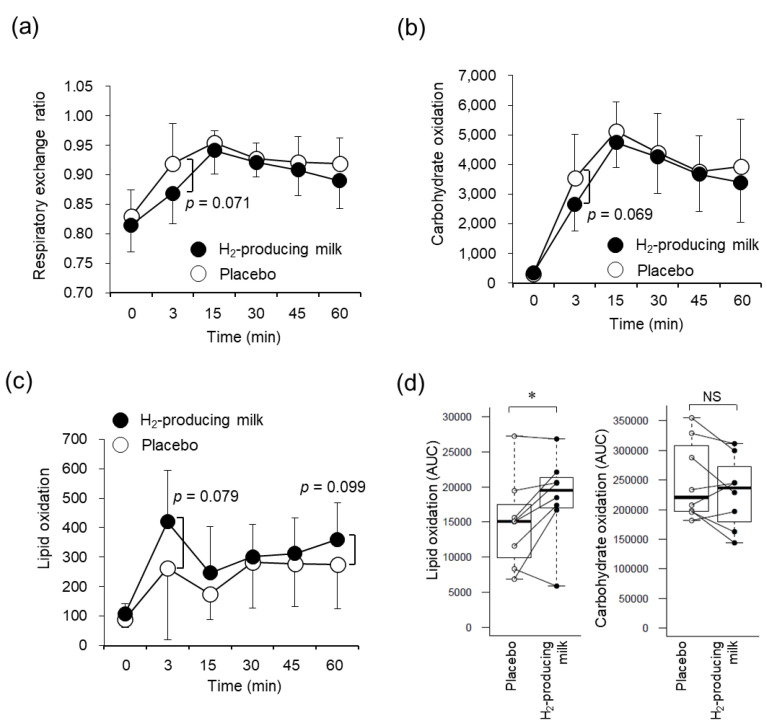
Effects of H_2_-producing milk and placebo use on energy metabolism during exercise. Time courses of the respiratory exchange ratio (**a**), lipid oxidation (**b**), and carbohydrate oxidation (**c**) during exercise are shown. Error bars represent the standard deviation of the mean. The black and white plots represent the means during the H_2_-producing milk and placebo trials, respectively. A comparison of the AUC for lipid oxidation and that for carbohydrate oxidation (**d**) is also shown (paired *t*-test, * *p* < 0.05). Each part of the boxplot diagram shows the following: upper line, maximum value; lower line, minimum value; box, interquartile range; bold line inside the box, median; and white and black plots, values during the placebo and H_2_-producing milk trials, respectively. AUC, area under the curve.

**Table 1 nutrients-14-04875-t001:** Visual analog scale scores during trial.

	Measurements						Interaction	Time	Trial
		Pre	Post	Post 30	Post 60	Next Morning	(η_p_^2^)	(η_p_^2^)	(η_p_^2^)
General muscle soreness	HPM	2.5 (4.5)	27.0 (28.3) ^a^	25.0 (22.3) ^a^	10.0 (22.5)	6.5 (16.0)	0.418	<0.001	0.113
	Placebo	3.0 (7.3)	41.5 (27.8) ^a^	29.0 (21.3) ^a^	18.5 (19.5)	24.5 (17.5) ^b^	(0.126)	(0.637)	(0.319)
Lower limbs soreness	HPM	3.0 (6.3)	34.5 (33.8) ^a^	27.0 (31.5)	11.0 (29.5)	12.5 (30.5)	0.351	<0.001	0.400
	Placebo	3.5 (6.5)	48.5 (40.3) ^a^	28.5 (27.0) ^a^	21.5 (37.0)	28.5 (32.8) ^a^	(0.142)	(0.646)	(0.103)
Fatigue	HPM	2.0 (9.0)	58.5 (22.0) ^b^	49.0 (26.3) ^a^	38.0 (38.0)	27.0 (28.0) ^a^	0.878	<0.001	0.680
	Placebo	12.0 (14.3)	67.0 (13.0) ^b^	43.0 (31.8)	38.0 (24.3)	38.5 (15.8) ^a^	(0.041)	(0.744)	(0.026)

All data are represented as median and (interquartile range). HPM: H2-producing milk, Pre: before exercise, Post 0: immediately after exercise, Post 30: 30 min after exercise, Post 60: 60 min after exercise, Next morning: the first morning after the trial. Significant differences are shown as follows: ^a^
*p* < 0.05, ^b^
*p* < 0.01 vs. Pre (Converted to the rank value and evaluated using two-way repeated measures analysis of variance and adjusted using the Bonferroni method).

## Data Availability

All sequence data of fecal microbiota are available in the DDBJ Sequence Read Archive database under the accession number DRA013790.

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
