# Peer review of "Effects of Intestinal Bacterial Hydrogen Gas Production on Muscle Recovery following Intense Exercise in Adult Men: A Pilot Study"

_nutrients, 2022, doi:10.3390/nu14224875_

Round 1
Reviewer 1 Report
Thank you for your submission to the Nutrients. This was an interesting paper addressing the use of a proprietary supplement that has little to no research on it to date. As such, any information surrounding its efficacy is notable and potentially useful. The paper was easy to follow with concepts, methods, results, and conclusions were clearly stated. The overall finding from this paper is that there is a possibility that the consumption of a H2-containing milk beverage may improve recovery following exercise.
There were some general concerns I had about the conclusions drawn. For the most part, I believe these can be addressed.
· The use of 8-OHdG as a marker for oxidative stress from an acute bout of exercise should be better defended/explained. This marker is better established as a marker of longitudinal health status. There are even some reports of no change in 8-OHdG following a fairly intense bout of exercise (1).
· As pointed out by the authors (lines 330-333), it is generally accepted that complete reduction of oxidative stress may prevent athletic development. The level of reduced ROS observed following H2 milk consumption is postulated to strike the right balance to allow for fitness gains and speed recovery. What is this based on?
· The observation of increased lipid oxidation without any difference in carbohydrate consumption (discussed in paragraph starting with line 334) was interesting. This would mean that more energy was used in the experimental condition than was with the control -a lower cycling economy. If actual calories expended were calculated from gas and lactate values, is there any difference observed?
· The assessment of “better recovery” following exercise is notoriously difficult. Use of the VAS’s for soreness and fatigue are common methods to capture this data along with some of the less subjective values in the serum assays included. Given the lack of difference in the objective measures, the VAS results are not strong indicators that differences came from actual the experimental/control variable. I do believe that proper care was taken in controlling diet, etc. surrounding the sessions. But, the point of expediting recovery is to enable the athlete to successfully perform the next training bout. Having added in a performance measure of some kind would have greatly helped in this regard. To be clear, I believe the authors did a good job controlling for extrinsic variation, and included both subjective and objective measures. The data simply did not paint a clear picture of enhanced recovery due solely to the H2 milk consumption -subjective markers plus relative changes in 8-OHdG.
· I think the discussion did well to note several limitations of the study. Still, given the lack of differences in serum and urine markers, I believe the level of confidence should be scaled back to reflect this. For example, changing “can likely” in line 375 to “may possibly” along with any other pertinent statements would be sufficient to address this.
· Given the use of an acute exercise bout in a group of individuals that were not-elite athletes, I think the introduction and discussion should have focused more on the acute response to exercise rather than on the response to some chronic load of sustained training, as might be seen in a situation of overreaching.
· Removal of subjects 5 and 9 due to higher H2 concentration with placebo than H2 milk consumption makes sense in the context of the idea that the H2 milk consumption is basically a vehicle for raising H2 levels. However, with the idea that the “product” being tested could potentially be marketed to athletes, if 20% of the population does not respond to said product (similar to the idea of responders and non-responders to caffeine), this should be included in the results. Were they removed before data collection or after. Given the double-blind design, I thought perhaps after. In that case, would it be possible to publish the results including those 2 individuals? Perhaps as supplementary data?
Some more minor concerns, specific weaknesses, or clarifications required:
· The use of non-athletes to draw conclusions about how this beverage might be used to benefit athletes is teleologically incongruous. Could actual VO2max values be reported to better reflect the sample in the participant pool?
· Please change “VO2max” in line 134 to “VO2”
1. Yasuda, N., Bolin, C., Cardozo-Pelaez, F., & Ruby, B. C. (2015). Effects of repeated bouts of long-duration endurance exercise on muscle and urinary levels of 8-hydroxy-2ʹ-deoxyguanosine in moderately trained cyclists. Journal of sports sciences, 33(16), 1692-1701.
Author Response
Thank you for providing the opportunity to improve our manuscript. We have revised the manuscript in accordance with the reviewers' comments, which we believe has significantly enriched the manuscript. Please read the uploaded file.

Reviewer 2 Report
This manuscript describes the effect of a compounded milk drink that produces breath hydrogen, which has been shown to improve exercise performance. The study is well-designed, and the investigators have used several measures to test the efficacy of the milk drink. Galacto-oligosaccharide, maltitol, and glucomannan have been added to the milk drink to produce hydrogen.
Ten subjects were recruited for the study, and eight completed it. How was the sample size determined? This needs to be described in the statistics section.
More descriptive information should be provided about the participants. For example, the authors state they were not athletes. What was the participant’s physical activity level?
The authors state that the hydrogen gas is intestinally derived, yet, do not provide evidence that this is the location of its origin. The investigators started collecting the breath hydrogen samples five hours after consumption of the milk. This would suggest that the hydrogen is generated in the stomach or upper part of the small intestine, not the ileum, jejunum, or colon.
The paper does not describe how the fecal sample was collected and when it was collected. More detail on this must be added.
The rationale for collecting the fecal sample is not clear. For example, what data suggests the breath hydrogen was derived from the colon, the site of the fecal sample?
The words “can likely” in the manuscript’s last sentence should be changed. The data do not show that the milk drink can prevent the overtraining syndrome.
Author Response
Thank you for providing the opportunity to improve our manuscript. We highly appreciate the insightful and valuable comments of the reviewer. We have revised the manuscript in accordance with the reviewers' comments, which we believe has significantly enriched the manuscript. Please read the uploaded file.
